# How Do Young and Old Spontaneously Hypertensive Rats Respond to Antihypertensive Therapy? Comparative Studies on the Effects of Combined Captopril and Nifedipine Treatment

**DOI:** 10.3390/biomedicines10123059

**Published:** 2022-11-28

**Authors:** Beate Rassler, Christina Hawlitschek, Julia Brendel, Heinz-Gerd Zimmer

**Affiliations:** Carl-Ludwig-Institute of Physiology, University of Leipzig, 04103 Leipzig, Germany

**Keywords:** young SHR, old SHR, antihypertensive therapy, combination therapy, treatment effect, systolic blood pressure, LV hypertrophy, cardiac fibrosis

## Abstract

Numerous studies on the effects of antihypertensive treatment in young spontaneously hypertensive rats (SHRs) have shown that early-onset therapy may effectively reduce their blood pressure (BP) even to normotensive values. In contrast, only a few studies investigated the effects of treatment started at an advanced age. These studies revealed that antihypertensive effects are lower in adult or even in senescent SHRs compared with young SHRs. Even more, prevention of cardiac sequelae of hypertension such as hypertrophy and fibrosis is less effective when treatment starts late in life. Because, in patients, combination therapies with calcium antagonists are favored, we studied the efficacy of a combination therapy with captopril and nifedipine in young and old SHRs. We directly compared the treatment effects on BP as well as on cardiac hypertrophy and remodeling between these two animal cohorts. With antihypertensive treatment, significantly lower BP values were achieved in young SHRs despite a shorter treatment period compared with old SHRs. Although treatment effects on cardiac hypertrophy were greater in old than in young SHRs, cardiac fibrosis was significantly attenuated only in young but not in old SHRs. The results emphasize the value of antihypertensive therapy and particularly accentuate the importance of an early-onset therapy. With respect to problems such as late diagnosis and poor therapy adherence, these results may have great importance for the treatment of human hypertension.

## 1. Introduction

With a global prevalence of about 40% in the adult population [1], arterial hypertension is the main risk factor for morbidity and mortality of cardiovascular diseases and is considered to be the leading cause of death in industrialized countries [2,3,4]. In the overwhelming majority of patients, hypertension manifests in adulthood. The number of hypertensive adults aged 30–79 years in the world is more than 1.2 billion people at present [5]. The prevalence of hypertension increases with advancing age to 60–70% in people aged over 60–65 years [3,6]. From all hypertensive patients, only about 50% are diagnosed, and in about 40–50% of diagnosed patients, treatment is lacking or insufficient to stabilize blood pressure (BP) at values below 140/90 mmHg [5,7,8,9]. Untreated or insufficiently treated arterial hypertension leads to constant pressure load of the left ventricle (LV), which first results in cardiac hypertrophy. With long maintenance of hypertension, it may progress to cardiac remodeling, fibrosis, dilatation, and finally heart failure. One of the main causes of lacking or insufficient treatment of diagnosed hypertension is poor adherence or even non-adherence to treatment [3]. The term adherence denotes the extent to which the patient complies with the therapy recommendation. Non-adherence to antihypertensive therapy affects 10–80% of all hypertensive patients, preferably elderly patients [9,10]. Poor adherence to antihypertensive treatment is associated with elevated BP values, an increased cardiovascular risk, and a poor prognosis in these patients [3,9].

Although the prevalence of hypertension is significantly higher in persons above 60 years, about 13% of younger adults between 20 and 40 years of age worldwide are hypertensive with an average BP above 140/90 mmHg [11]. In a study on young adults, only 11% of the hypertensive persons received antihypertensive medication at the time of enrolment [12]. Hypertension is less frequently diagnosed in young than in old adults [11]. Younger adults may be in their prehypertensive state, which may remain undetected or underestimated for a longer period of time. However, four out of five prehypertensive people aged 40–49 years would become hypertensive in the next 10 years [13]. Prehypertensive and even hypertensive young people are often not aware of their BP [14]. However, several studies have demonstrated that elevated BP at youth correlates with higher rate of LV hypertrophy and cardiovascular mortality at an older age [15,16,17]. Therefore, the ESC/ESH recommends that all adults ≥18 years should have their office BP measured and recorded, and be aware of their BP [3].

Several years ago, the SPRINT study was the first randomized trial that clearly demonstrated the benefits of a systolic BP (SBP) reduction below previous target values to 120 mmHg [18,19]. A change in BP of 20/10 mmHg is associated with a 50% difference in cardiovascular risk [9]. In patients who were at increased risk of cardiovascular disease, antihypertensive therapy targeting SBP below 120 mmHg resulted in lower rates of major adverse cardiovascular events and mortality compared with target SBP of less than 140 mmHg [20]. Consequently, the BP targets have been revised in the recent European, U.S., and WHO guidelines [3,21,22]. Early initiation of therapy and faster achievement of the recommended BP targets (within 6 months, ideally even within 3 months) are associated with a significant reduction in heart failure and other major cardiovascular events [23]. For initiation of antihypertensive therapy, a combination treatment of a blocker of the renin-angiotensin system (RAS; either an angiotensin-converting enzyme (ACE) inhibitor or an angiotensin receptor blocker) with a calcium channel blocker (CCB) or diuretic is the preferred recommendation for most hypertensive patients. Combination therapy, preferably as single-pill combination, has been proven to be more effective in fast achievement of the recommended BP targets and to be associated with a better adherence of patients to treatment [3,23].

### Purpose of the Present Study

To prevent the development of cardiovascular complications, adequate therapy should start as early as possible. In animal models, efficient antihypertensive therapy started in their prehypertensive state proved to be more efficient than the same therapy started in adulthood when hypertension was manifest [24,25]. In humans, there is still much controversy about the treatment of younger adults with low-risk grade 1 hypertension and about the efficacy of antihypertensive treatment in young hypertensives compared with old patients. Conventional clinical outcome studies are difficult to perform in people before age 40, hence there is a lack of randomized clinical trials for young hypertensive patients [3,11]. Consequently, it is not clear whether the effects of antihypertensive treatments in old and young people are comparable. Even in animal models, there are only very few studies comparing treatment effects in young and old animals. Previously, we had performed studies on antihypertensive treatments in young (aged 7–10 weeks) [26] and old (aged 60–82 weeks) spontaneously hypertensive rats (SHRs) [27]. The present study was designed to directly compare the treatment effects achieved in young and old SHRs. The relative reduction of SBP, parameters of LV hypertrophy and LV remodeling and fibrosis induced by a combination of the ACE inhibitor captopril and the CCB nifedipine was compared using a two-way ANOVA with the factors treatment and age. The purpose of this analysis is to show how effectively the reduction in SBP and attenuation of cardiac hypertrophy and remodeling can be achieved with early-onset compared with late-onset therapy.

## 2. Materials and Methods

We re-evaluated data from two previously published studies on male SHRs [26,27]. The raw data were obtained from these studies and were re-analyzed using a new statistical approach allowing a direct comparison between the data from both animal studies. All animal protocols were approved by the state agency (Landesdirektion Sachsen, number and date of approval: TVV 36/08; 13 May 2009) in accordance with the Guide for the Care and Use of Laboratory Animals published by the National Institutes of Health and with the “European Convention for the Protection of Vertebrate Animals used for Experimental and other Scientific Purposes”.

In brief, we analyzed data from 12 young SHRs aged 7–10 weeks (ySHRs) and fourteen old SHRs aged 60–82 weeks (oSHRs). The study on ySHRs [26] was designed as a pilot study to identify the most appropriate treatments to be applied to oSHRs. The ySHRs were intended to be in their adolescence throughout the experiment. For this reason, the experimental period was defined from 7 to 10 weeks of age. In contrast, the old SHRs should be treated over a long period of time, which was set to 22 weeks [27].

All animals were cultured at Charles River, Sulzfeld, Germany. Both groups were subdivided at random into two subgroups each, one remaining untreated and serving as control (yCtrl, n = 6; oCtrl, n = 7) and the other one being treated with a combination of captopril (60 mg kg^−1^ d^−1^, Axxora, Lörrach, Germany) and nifedipine (10 mg kg^−1^ d^−1^, Sigma-Aldrich Chemie, Steinheim, Germany). The treated groups were labeled as yC+N (n = 6) and oC+N (n = 7). ySHRs were 7 weeks of age at the beginning of the treatment period, which lasted 3 weeks. In oSHRs, the treatment period started at 60 weeks of age and lasted 22 weeks. The medication (or placebo in Ctrl animals) was administered as tablets for oral uptake given into the cages along with chow once daily between 9:00 a.m. and 10:00 a.m.

Before and during the experimental period, we regularly measured systolic blood pressure (SBP) using the tail-cuff-method (TSE Blood Pressure Monitor, Series 209002, TSE Systems GmbH, Bad Homburg, Germany; for more details, see [26,27]). Values from the final SBP measurement performed in the last experimental week were taken for the present re-evaluation.

At the end of the experiments, the animals were sacrificed and their hearts were extracted. We determined heart weight (HW) as a measure of cardiac hypertrophy. We analyzed the absolute HW as well as HW normalized to final body weight (HW/BW). Pieces of the left ventricle (LV) were obtained to perform ribonuclease protection assays to determine several markers of LV hypertrophy and remodeling. mRNA expressions of atrial natriuretic peptide (ANP) and collagen types I (Coll 1) and III (Coll 3) were included into the present analysis. Histological preparations were made from the cardiac apex. A score ranging from 0 (= no fibrosis) to 3 (= fibrosis of the entire heart) was applied to assess the histological degree of fibrosis (for more details, see [26,27]). The degree of fibrosis was also included in the present analysis.

Statistical analyses were carried out with the software package SigmaPlot Version 14.0 (Systat Software GmbH, Erkrath, Germany) for Windows. First, we directly compared final SBP, HW, HW/BW, ANP mRNA, Coll 1 mRNA, Coll 3 mRNA, and degree of fibrosis among all four groups (yCtrl, yC+N, oCtrl, oC+N) using a two-way ANOVA with the factors A = treatment (Ctrl or C+N) and B = age (y or o). As a post-hoc test, an all pairwise multiple comparison procedure according to the Holm–Sidak method was applied.

In addition, we assessed the treatment effect for each parameter and compared the treatment effects in young versus old SHRs. The treatment effect was calculated for each treated animal as the difference between the average value of the respective Ctrl group (y or o) and the individual value of the treated animal. The difference was normalized to the average yCtrl or oCtrl value (given in %). A positive value of the treatment effect marks an improvement (i.e., reduction) in the parameter. For the comparison of young versus old, we first performed a Shapiro–Wilk test for normality. In the case of normal distribution, we performed a t-test, otherwise a Mann–Whitney rank sum test was applied. *p*-values < 0.05 were considered significant.

## 3. Results

### 3.1. Treatment Effects on SBP

SBP of young SHRs at baseline (age 7 weeks) was 173.8 ± 3.1 and 157.5 ± 3.7 mmHg for Ctrl and C+N animals, respectively (Table 1). This difference was not significant. The original experiment [26] included a total of 42 ySHRs with an average SBP at baseline of 171.2 ± 3.5 mmHg. In untreated ySHRs, SBP increased continuously over the 3-week period of observation to 201.7 ± 6.5 mmHg at 10 weeks of age. This was significantly higher than SBP of age-matched SHRs after 3 weeks of C+N treatment. In the first two weeks of treatment, SBP decreased to 115.8 ± 2.4 mmHg. SBP of age-matched Wistar–Kyoto rats (WKY) was with 123.1 ± 0.9 mmHg in a similar range [26]. However, in the final treatment week, SBP re-increased to 152.5 ± 2.8 mmHg. The difference to untreated ySHRs was with 49.2 ± 2.8 mmHg significant (*p* < 0.001) corresponding to a treatment effect of 24.4 ± 1.4% of untreated Ctrls.

Age had a significant effect on SBP (*p* < 0.001). In untreated oSHRs aged 60 weeks, SBP was 231.0 ± 8.7 mmHg and hardly rose during the following 22 weeks to 239.0 ± 5.1 mmHg at week 82 of age (*p* < 0.001 compared with untreated ySHRs). Treatment with C+N over 22 weeks significantly reduced SBP by 45.4 ± 4.8 mmHg to 193.6 ± 4.8 mmHg (*p* < 0.001; Table 1); this is, however significantly higher than in yC+N rats (*p* < 0.001; Figure 1A). The treatment effect is 19.0 ± 2.0% of the untreated oCtrls. However, despite a much longer period of therapy in oSHRs, this treatment effect is lower than in ySHRs (*p* = 0.059; Figure 1B).

### 3.2. Treatment Effects on Cardiac Hypertrophy

#### 3.2.1. Heart Weight

The relative heart weight (HW/BW) of ySHRs was 3.50 ± 0.07 mg/g without therapy and 3.06 ± 0.06 mg/g (*p* = 0.331) after 3 weeks of C+N therapy, which is a treatment effect of 12.6 ± 1.6%. In untreated oSHRs, HW/BW was significantly higher (5.51 ± 0.51 mg/g, *p* < 0.001), but 22 weeks of C+N treatment significantly reduced HW/BW to 3.44 ± 0.20 mg/g (*p* < 0.001), which is in a similar range as HW/BW of yCtrl rats (Figure 2A). The treatment effect was with 37.4 ± 3.7%, significantly greater than in ySHRs (*p* < 0.005; Figure 2B).

However, the body weight (BW) of the animals changed in a different way; while BW in 10-week-old SHRs was similar in the untreated and treated groups (309.7 ± 7.3 and 308.0 ± 7.3 g, respectively), 82-week-old untreated SHRs had a considerably lower BW (377.6 ± 37.2 g) than age-matched C+N-treated rats (418.6 ± 5.4 g; *p* = 0.042; Table 1). Moreover, SHRs aged 7–10 weeks are still juvenile, and their HW/BW ratio is substantially different from that of old SHRs aged 60–82 weeks. Therefore, analysis of the absolute HW may provide more realistic information about the treatment effects (Figure 2C,D). In ySHRs, HW was 1095 ± 40 mg in untreated SHRs and 934 ± 28 mg in C+N-treated animals (*p* = 0.136), which means a treatment effect of 14.7 ± 2.6%. In 82-week-old untreated SHRs, HW was significantly higher than in young animals (*p* < 0.001). Twenty-two weeks of C+N treatment significantly decreased HW from 1860 ± 93 mg to 1440 ± 81 mg (*p* < 0.001) corresponding to a treatment effect of 26.8 ± 1.3%. Although this treatment effect was significantly greater than in ySHRs (*p* = 0.035), HW of treated oSHRs remained significantly higher than in ySHRs after only 3 weeks of treatment (*p* < 0.001).

To assess to what extent the treatment effect on HW is mediated by SBP reduction, we further calculated HW reduction per unit SBP reduction. While in yC+N rats, HW decreased by 3.5 ± 0.7 mg/mmHg, this ratio was significantly greater in oC+N animals (9.7 ± 2.1 mg/mmHg, *p* = 0.035; data not shown).

#### 3.2.2. ANP mRNA

ANP as a marker of ventricular hypertrophy in 10-week-old untreated SHRs was 13.3 ± 4.0. After 3 weeks of treatment, age-matched SHRs presented a value of 4.3 ± 0.8 (*p* = 0.775), which is a treatment effect of 67.8 ± 6.0%. There was a significant effect of age on ANP expression (*p* < 0.001): In 82-week-old untreated SHRs, it was 235.4 ± 43.8, and 22 weeks of treatment decreased it to 45.7 ± 10.9; *p* < 0.001; Figure 3A). The treatment effect was 80.6 ± 4.6%, which is in the same range as in ySHRs (*p* = 0.118; Figure 3B) despite a much longer period of treatment.

### 3.3. Treatment Effects on Cardiac Remodeling and Fibrosis

#### 3.3.1. mRNA Expression of Collagens I and III

With transition to fibrosis, expression of collagen mRNA can be observed. In untreated ySHRs, the mRNA expression of Coll 1 and Coll 3 was 13.9 ± 2.1 and 22.5 ± 1.9, respectively. Treatment had no significant effect; the mRNA expression of the two collagens was even slightly increased to 15.4 ± 3.4 and 23.4 ± 3.7, respectively.

In 82-week-old untreated SHRs, mRNA expression of Coll 1 and Coll 3 was in a similar range, with 15.9 ± 4.6 and 25.4 ± 4.2, respectively. However, 22 weeks of treatment significantly reduced the collagen mRNA expressions to 4.1 ± 0.8 (Coll 1; *p* = 0.007) and 9.4 ± 1.5 (Coll 3; *p* < 0.001). The treatment effects were significantly higher than in young animals (Coll 1: 74.5 ± 5.2%, *p* = 0.003; Coll 3: 63.1 ± 6.0%, *p* = 0.001; Figure 4).

#### 3.3.2. Fibrosis

Mild to moderate histological signs of fibrosis were already observed in 10-week-old untreated SHRs (fibrosis degree: 1.47 ± 0.12). Three weeks of treatment reduced the degree of fibrosis to 0.59 ± 0.17; *p* < 0.001. The treatment effect was 58.0 ± 11.9%.

Age had a significant effect on the degree of fibrosis. Untreated SHRs aged 82 weeks presented moderate to severe signs of fibrosis (fibrosis degree: 2.30 ± 0.11), which was significantly higher than in ySHRs (*p* < 0.001). After 22 weeks of treatment, there were still signs of moderate fibrosis (fibrosis degree: 2.13 ± 0.07) corresponding to a treatment effect of 7.5 ± 3.1%, which was significantly smaller than that in ySHRs (*p* < 0.001; Figure 5).

## 4. Discussion

It is widely accepted that antihypertensive treatment should start as early in life as possible [3]. Epidemiological studies in humans have shown that young people with high BP were at higher risk of LV hypertrophy and cardiovascular disease events compared with normotensive persons [15,16,17]. Animal studies have shown that early-onset treatment exerts better effects with respect to SBP and cardioprotection than a therapy started at an advanced age [24,25]. However, only few studies have directly compared the effects of antihypertensive treatment in young and senescent SHRs.

### 4.1. Effects on SBP

The present results showed that treatment effects on SBP were greater in ySHRs than in oSHRs, thus confirming the experience from earlier studies [24,25]. Early treatment of SHR with RAS antagonists is able to reduce SBP even to normotensive values [24,28]. However, in their studies, the treatment started in the prehypertensive stage of life, that is, at or before the age of 4 weeks, and lasted 6–8 weeks. At 7 weeks of age, SBP of untreated SHRs ranged between 154 and 187 mmHg [26] and increased to 210–220 mmHg by week 12–15 [29]. For comparison, at that age, SBP of normotensive WKY ranges between 115 and 137 mmHg [24,26]. With further aging up to senescence, SBP increases only slightly; in untreated SHRs aged 60–82 weeks, it was about 240 mmHg (see Table 1). In contrast, SBP of old WKY remained in the range of young WKY [24]. The present results show that the treatment effect significantly depends on age; despite a much shorter interval of therapy (3 weeks only) treatment with C+N between the 7th and 10th week of life reduced SBP even more than a 22-week therapy with the same medication between the 60th and 82nd week of life.

Antihypertensive treatment started early in life has not only stronger but also more persistent effects on SBP. In young SHRs aged 4–6 weeks, transient antihypertensive therapy over 4–6 weeks resulted in a significant SBP reduction, which remained until age 24–30 weeks. In contrast, if such a treatment was initiated during adulthood (between week 20 and 24), the effect on SBP was significantly lower or even completely abolished [25,30,31,32]. Antagonists of the RAS not only exert vasodilatory effects, thus directly decreasing BP. In addition, they prevent or attenuate structural changes of resistance vessels. This effect, however, is only present in adolescent and young adult SHRs up to about 20 weeks of age, and is most pronounced at 4–10 weeks of age [33]. Of note, vascular antihypertrophic effects of RAS antagonists were also demonstrated in both young and old WKY [24]. Nifedipine is also able to reduce vascular hypertrophy in SHRs [34]. Prevention of vascular hypertrophy may explain the more potent antihypertensive effect of early-onset treatment and the long-term maintenance of BP reduction even after withdrawal of the therapy.

The effect of age found in our study cannot be differentiated from an effect of SBP at treatment onset; while SBP of all ySHRs presented here was lower than 200 mmHg at baseline, all of the oSHRs had a baseline SBP above 200 mmHg [26,27]. There was a significant positive correlation between age and SBP at baseline (r = 0.83; *p* < 0.001). These results strongly suggest that greater treatment effects and even normotension might be achieved the earlier in life the treatment is started.

### 4.2. Cardioprotective Effects

The therapy effects were not confined to antihypertensive effects but also included cardioprotective effects. With C+N treatment, cardiac hypertrophy and remodeling were differentially attenuated in young and old SHRs.

#### 4.2.1. Effects on Cardiac Hypertrophy and Remodeling in Young SHRs

In genetic forms of hypertension such as in SHRs or in humans with essential hypertension, vascular hypertrophy, and consequently elevated vascular resistance, cause pressure load to the LV, which in turn induces hypertrophy and remodeling of the LV myocardium. In a previous study on 6-month-old SHRs, both total peripheral resistance (TPR) and HW/BW were significantly higher than in age-matched WKY rats [35]. A similar observation has been made in the ySHR study: TPR of yCtrl rats was also significantly higher than in age matched WKY rats (0.20 ± 0.01 and 0.12 ± 0.02 mmHg·min·kg·mL^−1^, respectively; *p* = 0.01; unpublished data).

In SHR, development of cardiac hypertrophy starts after the prehypertensive stage between the 4th and 12th week of life [36,37]. At this stage, HW/BW of SHRs is about 13% higher than that of age-matched WKY [24]. Several studies showed that antihypertensive therapy with RAS antagonists started between the 4th and 14th week of life reduced HW/BW or LVW/BW by 19–27% [24,29,30,38,39]. Our study on young SHRs provided similar results: three weeks of C+N treatment reduced HW/BW by about 14%, but not to the level of normotensive WKY rats [26]. Without treatment, cardiac hypertrophy progresses over time, but even when treatment with RAS antagonists is started in adulthood (between week 24 and 34 of life), significant attenuation of cardiac hypertrophy can be achieved [24,40].

However, with progress of cardiac hypertrophy, remodeling processes including profibrotic processes will develop. Perrucci and co-workers demonstrated cardiac fibrosis and significantly increased collagen deposition in the hearts of 8-week-old untreated SHRs [41]. The progredient increase in blood pressure and the development of LV hypertrophy and fibrosis are accompanied by a deterioration of LV function. Echocardiographic examination demonstrated significant systolic and diastolic dysfunction in untreated SHRs even at 2–3 months of age [37]. With enalapril treatment over 14 weeks, deterioration of ejection fraction and fractional shortening as well as development of cardiac fibrosis were prevented or at least attenuated [42]. This is in line with our observations on ySHRs; at 10 weeks of age, they presented mild cardiac fibrosis, which was significantly reduced by more than 50% with three weeks of C+N treatment [26].

#### 4.2.2. Effects on Cardiac Hypertrophy and Remodeling in Old SHRs

With progress of age, the HW/BW differences between SHRs and WKY increased up to 29% at week 83 [24]. In the advanced stage of hypertension at week 82, cardiac hypertrophy of our untreated oSHRs had achieved a markedly higher degree than in young animals, as reflected by the significantly higher values of ANP mRNA expression (20 times higher than in yCtrls) and HW/BW (more than 30% higher than in yCtrls). Angiotensin II exerts many prohypertrophic effects on the heart. Consequently, prevention or attenuation of cardiac hypertrophy and transition into heart failure is a major goal of ACE inhibitor therapy [43]. Even in normotensive WKY rats, RAS antagonists can significantly reduce HW/BW [24]. In senescent SHRs at more than 80 weeks of age, short-term treatment with RAS antagonists has only weak effects: after 8 weeks of losartan therapy, HW/BW was only reduced by 7% [24]. In contrast, chronic captopril treatment administered from 14 to 24 months of age-reduced LVW/BW by about 30% [44], which is similar to the results of the present study. Direct antihypertrophic effects on cardiomyocytes have also been demonstrated for nifedipine [45]. Thus, cardioprotective effects of C+N treatment are not only based on antihypertensive effects, but in addition on direct antihypertrophic effects. This is also reflected in the decrease in HW related to the decrease in SBP, which was even higher in oSHRs than in ySHRs.

As cardiac hypertrophy progresses, remodeling and accumulation of extracellular matrix (ECM) advance too. Increased mRNA levels of ECM molecules such as transforming growth factor-beta and tissue inhibitor of metalloproteinases 2 have been found in 82-week-old untreated SHRs [27]. Of note, the levels of collagen mRNA and protein develop with aging in a nonsynchronous way: in young rats, collagen mRNA in the heart is relatively high, but does not result in collagen accumulation. In contrast, collagen mRNA in hearts of old rats is hardly elevated, but induces a two-fold increase in collagen protein content [46]. This may explain the similar levels of collagen mRNA in old and young untreated SHRs in the present study, which were associated with different histological degrees of fibrosis. While C+N treatment in ySHRs did not decrease collagen mRNA expression, but significantly reduced fibrosis, the opposite effect was achieved in oSHRs. Analysis of mRNA expression can only reflect a moment within a process developing over a longer period of time. In particular, accumulation of collagen is not linearly related to elevated levels of collagen mRNA. Hence, we consider the histological degree of fibrosis to be the more meaningful parameter with respect to cardiac remodeling. Even though the treatment effect on collagen mRNA was much greater in oSHRs compared with ySHRs, attenuation of fibrosis was only weak as the intervention was initiated in an advanced stage of cardiac remodeling. An early start of antihypertensive treatment may attenuate and delay the process of remodeling more effectively.

The transition into fibrosis is associated with further functional deterioration and finally leads to cardiac failure. At 80 weeks of age, diastolic function and compliance of the LV in untreated SHRs were impaired compared with age-matched WKY [47]. Treatment significantly improved ejection fraction index, but not to the level of normotensive WKY [44]. An echocardiographic study on old SHRs demonstrated a deterioration of both systolic and diastolic LV function between week 60 and 82 without treatment. Moreover, LV catheterization under anesthesia revealed signs of cardiac failure and LV decompensation. All of these changes were attenuated and delayed, but not completely prevented by antihypertensive treatment [48]. These results are in line with the weak antifibrotic effect of a late-onset treatment.

#### 4.2.3. Early-Onset versus Late-Onset Treatment

Similar to its effects on SBP, early-onset antihypertensive treatment also has long-lasting effects on cardiac hypertrophy and remodeling. Antagonization of prohypertrophic Angiotensin II effects in a sensitive period of life (up to age 20 weeks) makes even a transient treatment with ACE inhibitors effective with regard to prevention of LV hypertrophy [29,39].

The duration of treatment is an important factor with regard to the treatment effect. This is well illustrated by a comparison of our results with those of Demirci and co-workers: They applied losartan at the same dose over the same period of time to young and old SHRs and observed a lower treatment effect in the old animals [24]. In contrast, we administered the same medication (C+N) to young and old SHRs, but a more than seven-fold treatment interval was necessary to achieve a similar antihypertrophic treatment effect in oSHRs [26,27]. Lifelong antihypertensive treatment started at one month of age doubled the lifespan of stroke-prone SHRs to 30 months, which is identical to that of normotensive WKY. In contrast, 80% of untreated SHRs had died after 15 months [49]. These findings emphasize the importance of a long-lasting antihypertensive therapy, and this is the more important the later in life treatment is initiated.

#### 4.2.4. Applications to Antihypertensive Treatment in Humans

Our results on animals clearly show the advantages of an early start of an effective antihypertensive therapy, in particular with respect to preventing cardiac remodeling and thus transition into cardiac failure. This emphasizes the importance of starting BP control during early adulthood. Important aspects of antihypertensive treatment in humans are fast achievement of target BP values and adequate persistence and adherence to therapy. A combination of two or three drugs from different drug classes facilitates the adjustment of treatment to the individual patient as it allows dose reduction of the single drugs and minimization of adverse drug effects, thus ensuring the patient’s adherence to therapy. Consequently, a combination of two or more, ideally as fixed-dose single-pill administration, as initial therapy of hypertension is currently recommended in most guidelines [4,22].

### 4.3. Limitations of the Study

There are several limitations to this study. An important limitation is the lack of functional measurements and investigation of vascular changes. The animal studies were designed to analyze cardiac but not vascular sequelae of hypertension. Unfortunately, only systolic BP values were measured. We are aware of the fact that diastolic BP would provide relevant data characterizing the vascular treatment effects. This limitation is because of the method of non-invasive BP measurement with a tail cuff device that did not allow measurement of diastolic BP. Data on TPR, which may reflect vascular changes, were only obtained from the ySHRs. In addition, we have compared changes in cardiac tissue with echocardiographic data from oSHRs reported previously [48].

Another limitation is in the analysis of collagen mRNA expression to characterize the development of fibrosis. Previous studies on rats stimulated with norepinephrine showed a good correlation between LV collagen mRNA expression and collagen amount in the histological specimen [50,51]. Fibrotic processes include enhanced collagen turnover and an increased collagen accumulation. We assume that mRNA expression does not adequately reflect the accumulation of collagen. Moreover, collagen mRNA and protein content develop with aging in a nonsynchronous way [46]. Collagen protein concentration or hydroxyproline content would more adequately reflect the intensity of fibrotic processes. We consider the histological degree of fibrosis to be a reliable and meaningful marker of fibrosis.

## 5. Conclusions

As elevated BP is the leading cause of premature death and is the major risk factor for a variety of cardiovascular disease events [3], early and effective antihypertensive therapy is of paramount importance for the patients. The present data of an animal study show that even a late-onset treatment of hypertension has both antihypertensive and cardioprotective effects. However, in those animals, SBP remained at significantly higher levels than in ySHRs, which received the same medication administered over a much shorter period of time at an early stage of life. Consequently, cardiac remodeling and transition into fibrosis were delayed but not prevented in the old animals. In contrast, fibrosis was significantly attenuated in ySHRs after only three weeks of therapy. This result gives strong reason to believe that a life-long antihypertensive therapy started at an early stage in life may prevent severe cardiac remodeling and transition into cardiac failure at advanced age.

The results emphasize that antihypertensive therapy should start as early as possible and be maintained as long as BP is elevated, usually life-long. Continuous BP control and strict adherence to treatment are inevitable. Early reduction in SBP, preferably to normotensive or near-normotensive values, contributes to prevent cardiac complications of hypertension.

## Figures and Tables

**Figure 1 biomedicines-10-03059-f001:**
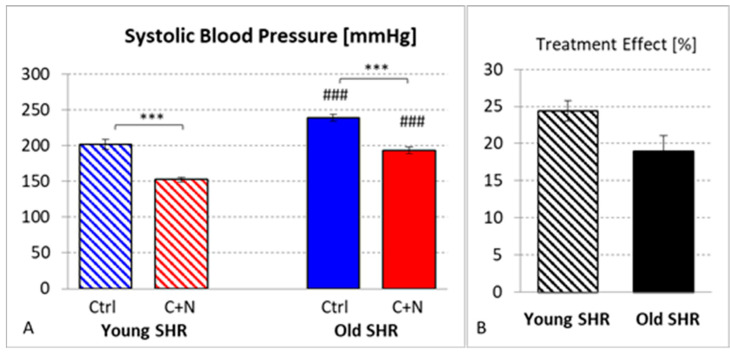
(**A**) Systolic blood pressure (in mmHg) at 10 (young SHR) or 82 (old SHR) weeks of age in untreated (Ctrl) and treated (captopril + nifedipine, C+N) SHRs. (**B**) Treatment effects in young and old SHRs (in % of corresponding average Ctrl value). Significance marks: ⌈*⌉ significant difference between marked groups, *** *p* < 0.001; ### significant difference to corresponding young SHR group, *p* < 0.001.

**Figure 2 biomedicines-10-03059-f002:**
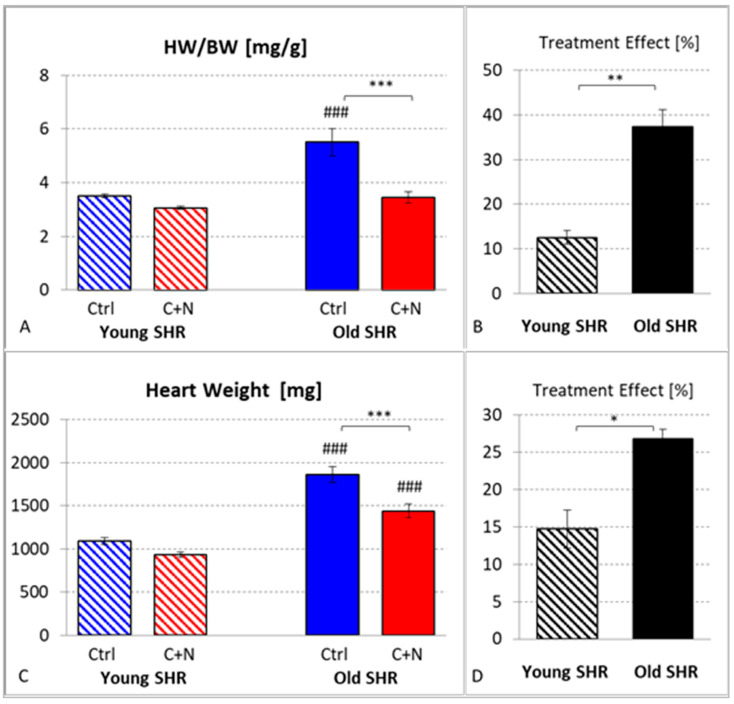
(**A**) Heart weight related to body weight in untreated (Ctrl) and treated (C+N) young and old SHRs. (**B**) Treatment effects on HW/BW in young and old SHRs (in % of corresponding average Ctrl value). (**C**) Absolute heart weight in untreated (Ctrl) and treated (C+N) young and old SHRs. (**D**): Treatment effects on heart weight in young and old SHRs (in % of corresponding average Ctrl value). Significance marks: ⌈*⌉ significant difference between marked groups, * *p* < 0.05, ** *p* < 0.01, *** *p* < 0.001; ### significant difference to corresponding young SHR group, *p* < 0.001.

**Figure 3 biomedicines-10-03059-f003:**
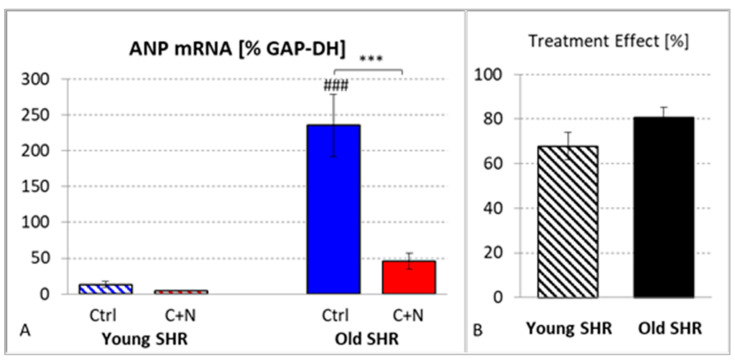
(**A**) mRNA expression of atrial natriuretic peptide in untreated (Ctrl) and treated (C+N) young and old SHRs. (**B**) Treatment effects in young and old SHRs (in % of corresponding average Ctrl value). Significance marks: ⌈*⌉ significant difference between marked groups, *** *p* < 0.001; ### significant difference to corresponding young SHR group, *p* < 0.001.

**Figure 4 biomedicines-10-03059-f004:**
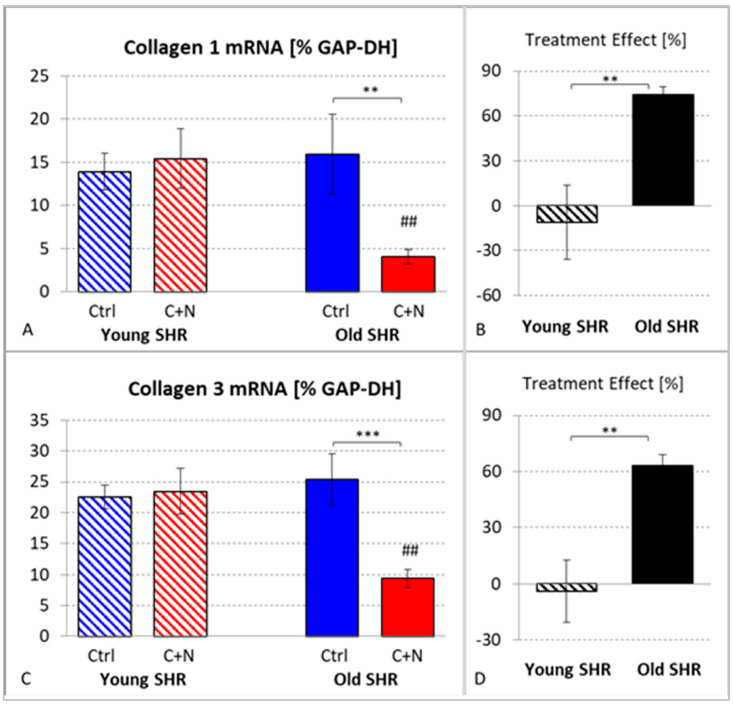
(**A**) mRNA expression of collagen type I in untreated (Ctrl) and treated (C+N) young and old SHRs. (**B**) Treatment effects on Coll 1 in young and old SHRs (in % of corresponding average Ctrl value). (**C**) mRNA expression of collagen type III in untreated (Ctrl) and treated (C+N) young and old SHRs. (**D**) Treatment effects on Coll 3 in young and old SHRs (in % of corresponding average Ctrl value). Significance marks: ⌈*⌉ significant difference between marked groups, ** *p* < 0.01, *** *p* < 0.001; ## significant difference to corresponding young SHR group, *p* < 0.01.

**Figure 5 biomedicines-10-03059-f005:**
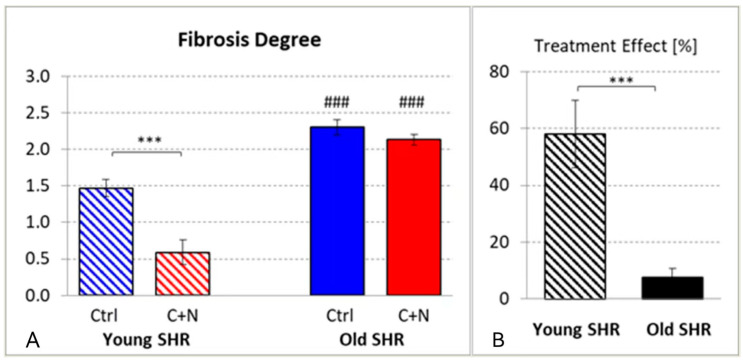
(**A**) Degree of fibrosis in untreated (Ctrl) and treated (C+N) young and old SHRs. (**B**) Treatment effects in young and old SHRs (in % of corresponding average Ctrl value). Significance marks: ⌈*⌉ significant difference between marked groups, *** *p* < 0.001; ### significant difference to corresponding young SHR group, *p* < 0.001.

**Table 1 biomedicines-10-03059-t001:** SBP, HR, and BW of young and old SHRs.

Animal Group	SBP [mmHg]		HR [min^−1^]		BW [g]	
	Baseline	Final	Baseline	Final	Baseline	Final
yCtrl	173.8 ± 3.1	201.7 ± 6.5 °	384.2 ± 10.3	416.5 ± 13.5	185.7 ± 4.2	309.7 ± 7.3 °°
yC+N	157.5 ± 3.7	152.5 ± 2.8	396.9 ± 9.1	397.7 ± 9.2	175.8 ± 7.3	308.0 ± 7.3 °°
oCtrl	231.0 ± 8.7	239.0 ± 5.1	413.5 ± 14.3	400.2 ± 20.6	396.7 ± 9.7	377.6 ± 37.2
oC+N	244.5 ± 8.1	193.6 ± 4.8 °	401.0 ± 14.6	409.2 ± 6.7	414.7 ± 8.6	418.6 ± 5.4 *

Data are given as means ± SEM. Baseline values were obtained at 7 weeks of age in young SHRs and at 60 weeks of age in old SHRs. Final values were obtained at 10 weeks of age in young SHRs and at 82 weeks of age in old SHRs. Treatment groups: y/oCtrl: untreated animals, y/oC+N: animals treated with captopril + nifedipine. * significant vs. corresponding Ctrl group (*p* < 0.05); ° significant vs. baseline value (*p* < 0.01); °° significant vs. baseline value (*p* < 0.001).

## Data Availability

Not applicable.

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
