# Peer review of "How Do Young and Old Spontaneously Hypertensive Rats Respond to Antihypertensive Therapy? Comparative Studies on the Effects of Combined Captopril and Nifedipine Treatment"

_biomedicines, 2022, doi:10.3390/biomedicines10123059_

Round 1
Reviewer 1 Report
This is an interesting study focused on the effects of early and late treatment of genetic hypertension in SHR by a combined antihypertensive therapy (captopril + nifedipine). The authors focused their attention on the changes in blood pressure, cardiac hypertrophy and fibrosis. The results seem to be realistic and their interpretation correct, but their discussion might be modified on the basis of following comments
1) It is a pity that the authors did not examine vascular changes in their experimental animals. It is well known that hypertension is caused by the elevation of systemic resistance but not by the changes in cardiac output (see Figs. 2 and 3 in Zicha and Kuneš Physiological Reviews 79: 1227, 1999). Therefore cardiac hypertrophy and/or fibrosis reflect rather the consequences of high blood pressure but not its cause. If there would be similar changes in resistance vessels as in the heart, the interpretation of the results would be easier (e.g. lines 278-283 or 321-335).
2) A comparison of cardiovascular effects elicited by early and late antihypertensive therapy with ACE/AT1 blockers was summarized by Lundie MJ et al. J Hypertens 15: 339, 1997 who paid the attention to cardiovascular structural changes including blood vessels. A considerable overview of age-dependent effects of antihypertensive therapy in SHR has been described by Zicha and Kuneš Physiol Rev 79: 1227, 1999 – see pages 1244-1249).
3) Please, note that Ref. 34 (Hojna et al 2007) is not a good example for significantly lower or abolished effect of RAS inhibition in adult SHR on their blood pressure (lines 275-277) because both blood pressure and heart weight were substantially reduced by the late captopril treatment. More appropriate data can be found in other paper of this group (Zicha et al. Physiol Res 63: 13, 2014 (see Figs 1, 3 and 4).
4) Ref. 29 (Racasan et al. 2004) should be used with a caution because there was used a very early RAS inhibition (last week of pregnancy + first few weeks before weaning). In this period RAS inhibition is known to damage the development of renal function.
5) MY MAJOR COMMENT concerns the most important finding of the study, i.e. cardiac fibrosis which was reduced only in young but not in old SHR, although the late therapy lasted 22 weeks (seven-fold longer compared to early treatment of young rats). It is interesting that the early short-term antihypertensive therapy did not lower heart weight but it reduced cardiac fibrosis in young SHR. Does it mean that cardiac fibrosis in SHR develops earlier than cardiac hypertrophy?
6) The reverse finding on the effects of the late long-term antihypertensive therapy in old SHR (reduction of cardiac hypertrophy but no change in fibrosis degree) is also interesting. Would you agree with Folkow´s statement that antihypertensive therapy reverses faster muscular than fibrotic changes? However, his statement was made on the basis of vascular changes in SHR treated with beta-blockers.
7) Finally, there is a considerable discrepancy between the effects of antihypertensive therapy on collagen 1/3 mRNA expression (reduction in old but not in young rats) and cardiac fibrosis (reduction in young but not in old rats). Maybe that old fashioned measurement of total collagen content (based upon hydroxyl-proline determination) would be more informative that modern mRNA expression of selected collagen types. In addition, both changes in translation and collagen metabolism might be responsible for this discrepancy. Could you comment this issue in more details.
Author Response
We thank this reviewer for his/her valuable and constructive comments on our manuscript.
1) It is a pity that the authors did not examine vascular changes in their experimental animals. It is well known that hypertension is caused by the elevation of systemic resistance but not by the changes in cardiac output (see Figs. 2 and 3 in Zicha and Kuneš Physiological Reviews 79: 1227, 1999). Therefore cardiac hypertrophy and/or fibrosis reflect rather the consequences of high blood pressure but not its cause. If there would be similar changes in resistance vessels as in the heart, the interpretation of the results would be easier (e.g. lines 278-283 or 321-335).
RESPONSE: We agree with this reviewer that cardiac hypertrophy and fibrosis are consequences of high blood pressure, not its cause. We have inserted a statement into the Discussion section (subsection 4.2.1. Effects on cardiac hypertrophy and remodeling in young SHR):
“In genetic forms of hypertension such as in SHR or in humans with essential hypertension, vascular hypertrophy and consequently, elevated vascular resistance cause pressure load to the LV, which in turn induces hypertrophy and remodeling of the LV myocardium. In a previous study on 6-month-old SHRs, both total peripheral resistance (TPR) and HW/BW were significantly higher than in age-matched WKY rats (Ziegelhöffer et al., 2006). A similar observation has been made in the ySHR study: TPR of yCtrl rats was also significantly higher than in age matched WKY rats (0.20 ± 0.01 and 0.12 ± 0.02 mmHg*min*kg/ml, respectively; p=0.01; unpublished data).”
Unfortunately, we have not investigated vascular changes. We have added a Limitations subsection and stated this as an important limitation to this study.
2) A comparison of cardiovascular effects elicited by early and late antihypertensive therapy with ACE/AT1 blockers was summarized by Lundie MJ et al. J Hypertens 15: 339, 1997 who paid the attention to cardiovascular structural changes including blood vessels. A considerable overview of age-dependent effects of antihypertensive therapy in SHR has been described by Zicha and Kuneš Physiol Rev 79: 1227, 1999 – see pages 1244-1249).
RESPONSE: We thank this reviewer for his/her valuable comment. We have inserted some statements into the Discussion section (subsection 4.1.Effects on SBP, 2nd paragraph):
“Antagonists of the RAS not only exert vasodilatory effects, thus directly decreasing BP. In addition, they prevent or attenuate structural changes of resistance vessels. This effect, however, is only present in adolescent and young adult SHR up to about 20 weeks of age, and is most pronounced at 4-10 weeks of age (Zicha & Kunes, 1999). … Prevention of vascular hypertrophy may explain the more potent antihypertensive effect of early-onset treatment and the long-term maintenance of BP reduction even after withdrawal of the therapy.”
3) Please, note that Ref. 34 (Hojna et al 2007) is not a good example for significantly lower or abolished effect of RAS inhibition in adult SHR on their blood pressure (lines 275-277) because both blood pressure and heart weight were substantially reduced by the late captopril treatment. More appropriate data can be found in other paper of this group (Zicha et al. Physiol Res 63: 13, 2014 (see Figs 1, 3 and 4).
RESPONSE: We thank this reviewer for this hint. We have replaced the reference Hojna et al 2007 in this part of the text (original lines 275-277) by the reference proposed by this reviewer (Zicha et al. 2014).
4) Ref. 29 (Racasan et al. 2004) should be used with a caution because there was used a very early RAS inhibition (last week of pregnancy + first few weeks before weaning). In this period RAS inhibition is known to damage the development of renal function.
RESPONSE: We thank this reviewer for this hint. We have deleted this reference.
5) MY MAJOR COMMENT concerns the most important finding of the study, i.e. cardiac fibrosis which was reduced only in young but not in old SHR, although the late therapy lasted 22 weeks (seven-fold longer compared to early treatment of young rats). It is interesting that the early short-term antihypertensive therapy did not lower heart weight but it reduced cardiac fibrosis in young SHR. Does it mean that cardiac fibrosis in SHR develops earlier than cardiac hypertrophy?
RESPONSE: We do not think that cardiac fibrosis develops earlier than cardiac hypertrophy. We think that cardiac remodeling starts later than cardiac hypertrophy. But both processes develop and progress in parallel over a long period of time.
In young SHR, C+N therapy also decreased HW significantly as has been reported in Hawlitschek et al., 2022, SJBS (reference #26). Of note, in the original analysis, we compared HW/BW of 5 animal groups (1 WKY, 1 untreated SHR group, 3 treated SHR groups). Their HW/BW values ranged between 2.7 and 3.5 mg/g, and HW/BW of the C+N animals was significantly lower than in the Ctrl group. In the present study, we compared these values with a value in a significantly higher range (HW/BW in oCtrls = 5.5 mg/g), and this may be the reason why HW/BW in yC+N was not significantly lower than in yCtrl. In old SHR, C+N treatment proved to be highly effective in reducing HW even at this stage of life. In contrast, fibrosis is characterized by increased collagen accumulation, which is less sensitive to treatment and hence, can hardly be reversed by treatment.
6) The reverse finding on the effects of the late long-term antihypertensive therapy in old SHR (reduction of cardiac hypertrophy but no change in fibrosis degree) is also interesting. Would you agree with Folkow´s statement that antihypertensive therapy reverses faster muscular than fibrotic changes? However, his statement was made on the basis of vascular changes in SHR treated with beta-blockers.
RESPONSE: We fully agree with this statement because we think that the beginning of muscular hypertrophy precedes the start of fibrosis. In particular, when compensated cardiac hypertrophy passes into heart failure, myocytes decay by necrotic and apoptotic mechanisms and trigger fibrotic processes and accumulation of extracellular matrix (Boluyt and Bing 2000). As we have not analyzed functional changes and thus, have no clear hint on beginning heart failure, we have not inserted this statement in our discussion. However, an echocardiographic study on oSHRs revealed signs of transition into heart failure in untreated but not in treated animals (Zimmer et al. 2015, referenced #48). This has been discussed in our paper.
7) Finally, there is a considerable discrepancy between the effects of antihypertensive therapy on collagen 1/3 mRNA expression (reduction in old but not in young rats) and cardiac fibrosis (reduction in young but not in old rats). Maybe that old fashioned measurement of total collagen content (based upon hydroxyl-proline determination) would be more informative that modern mRNA expression of selected collagen types. In addition, both changes in translation and collagen metabolism might be responsible for this discrepancy. Could you comment this issue in more details.
RESPONSE: We agree with this reviewer that mRNA expression may not adequately reflect the state of fibrotic processes. Previous studies from our group showed a good correlation between collagen mRNA expression and collagen amount in the histological specimen (Briest et al., 2001, 2003, references #50,51). However, collagen mRNA and protein content develop with aging in a nonsynchronous way [Besse et al., 1994, reference #46]. Fibrotic processes include enhanced collagen turnover and an increased collagen accumulation. Hence, hydroxyproline content would be more sensitive for detection of these processes. In our SHR studies, we also have used two different indicators of fibrosis. As the second (and more important) parameter, we have chosen the histological evidence of collagen accumulation as sign of cardiac fibrosis. We consider the histological degree of fibrosis to be a reliable and meaningful marker of fibrosis.
We have inserted some additional text into the Discussion section (subsection 4.2.2. Effects on cardiac hypertrophy and remodeling in old SHR):
“Of note, the levels of collagen mRNA and protein develop with aging in a nonsynchronous way: In young rats, collagen mRNA in the heart is relatively high but does not result in collagen accumulation. In contrast, collagen mRNA in hearts of old rats is hardly elevated but induces a two-fold increase in collagen protein content [Besse et al., 1994]. This may explain the similar levels of collagen mRNA in old and young untreated SHR in the present study, which were associated with different histological degrees of fibrosis. While C+N treatment in ySHR did not decrease collagen mRNA expression but significantly reduced fibrosis, the opposite effect was achieved in oSHR.”
In addition, we have inserted a Limitations section and included a statement there.
Reviewer 2 Report
The publication by Rassler et al. addresses the effect of antihypertensive therapy in young and adult SHR rats. The authors emphasize that the effect of therapy is significantly greater in young than in adult animals. This is certainly nothing revolutionary, as it is known that young animals are significantly more sensitive to antihypertensive therapy, not only to the one used in this study (captopril in combination with nifedipine), but also to other therapies used. I have several comments and questions about the work.
1) It is incomprehensible that ySHRs were treated for only 3 weeks, while oSHRs were treated for 22 weeks. Do the authors have any logical explanation for this difference? This may be related to the fact that ySHR are much more sensitive to antihypertensive therapy, as has been shown many times.
2) I believe that the determination of mRNA belongs to modern methodologies, but I am convinced that mRNA does not always reflect the real state. Determination of hydroxyproline concentration could better reflect the actual state of cardiac muscle hypertrophy (eg Conrad et al., Circulation 1995, Matsuhara et al., Am J Physiol Heart Circ Physiol 2000 etc.)
3) What was the effect of combination therapy on BW? If the difference in HW reduction between oSHR and ySHR was more than 15% and the difference in HW/BW was almost 30%, could this be a change in BW?
4) What type of staining was used for staining in determining fibrosis?
If I can speculate with the authors. The statement on lines 281-283 is certainly true, as evidenced primarily by Harrap's study. However, how can the results of animal studies be applied to humans? How to determine the prehypertensive stage in patients? Etc.
Author Response
We thank this reviewer for his/her valuable comments.
- It is incomprehensible that ySHRs were treated for only 3 weeks, while oSHRs were treated for 22 weeks. Do the authors have any logical explanation for this difference?This may be related to the fact that ySHR are much more sensitive to antihypertensive therapy, as has been shown many times.
RESPONSE: The study on ySHRs was designed as a pilot study to identify the most appropriate treatment regimens that should be applied to oSHRs. The young rats were intended to be in their adolescence throughout the experiment. Adolescence in male rats lasts until about 10 weeks of age (Sengupta P, Int J Prev Med 2013,624-630). In contrast, the old SHRs should be treated over a long period of time, which was set to 22 weeks. We inserted this statement into the Methods Section.
- I believe that the determination of mRNA belongs to modern methodologies, but I am convinced that mRNA does not always reflect the real state. Determination of hydroxyproline concentration could better reflect the actual state of cardiac muscle hypertrophy (eg Conrad et al., Circulation 1995, Matsuhara et al., Am J Physiol Heart Circ Physiol 2000 etc.)
RESPONSE: We agree with this reviewer that mRNA expression may not adequately reflect the state of fibrotic processes. Previous studies from our group showed a good correlation between collagen mRNA expression and collagen amount in the histological specimen (Briest et al., 2001, 2003, references #50,51). However, collagen mRNA and protein content develop with aging in a nonsynchronous way [Besse et al., 1994, reference #46]. Fibrotic processes include enhanced collagen turnover and an increased collagen accumulation. Hence, hydroxyproline content would be more sensitive for detection of these processes. In our SHR studies, we also have used two different indicators of fibrosis. As the second (and more important) parameter, we have chosen the histological evidence of collagen accumulation as sign of cardiac fibrosis. We consider the histological degree of fibrosis to be a reliable and meaningful marker of fibrosis.
We have inserted some additional text into the Discussion section (subsection 4.2.2. Effects on cardiac hypertrophy and remodeling in old SHR):
“Of note, the levels of collagen mRNA and protein develop with aging in a nonsynchronous way: In young rats, collagen mRNA in the heart is relatively high but does not result in collagen accumulation. In contrast, collagen mRNA in hearts of old rats is hardly elevated but induces a two-fold increase in collagen protein content [Besse et al., 1994]. This may explain the similar levels of collagen mRNA in old and young untreated SHR in the present study, which were associated with different histological degrees of fibrosis. While C+N treatment in ySHR did not decrease collagen mRNA expression but significantly reduced fibrosis, the opposite effect was achieved in oSHR.”
In addition, we have inserted a Limitations section and included a statement there.
- What was the effect of combination therapy on BW? If the difference in HW reduction between oSHR and ySHR was more than 15% and the difference in HW/BW was almost 30%, could this be a change in BW?
RESPONSE: We have added a Table showing pre- and post-treatment values of SBP, HR and BW from all 4 groups. BW of oC+N rats did not change over the 22 weeks of treatment. We think that at least part of the increase in HW/BW of oCtrl rats (and hence, of the treatment effect in oSHRs) is due to a decrease in the BW of oCtrls; therefore, we have additionally shown the changes in absolute HW. With this parameter, the difference in the treatment effect between ySHR and oSHR is smaller than with HW/BW, but this reflects the effect more adequately.
- What type of staining was used for staining in determining fibrosis?
RESPONSE: We have used Masson’s trichrome staining. This has been described in the original papers from Hawlitschek et al., SJBS 2022,ref #26, and Hawlitschek et al., Biomedicines 2022, ref #27). Masson’s trichrome staining is appropriate to indicate collagen fibres (stained in blue).
- If I can speculate with the authors. The statement on lines 281-283 is certainly true, as evidenced primarily by Harrap's study. However, how can the results of animal studies be applied to humans? How to determine the prehypertensive stage in patients? Etc.
RESPONSE: Determination of a prehypertensive stage in humans is probably very difficult as it would need a longitudinal study starting in the childhood and accompanying the persons until adulthood and even longer. The sentence on lines 281-283 was primarily meant to relate to animals, but it was probably misleading in its original form. We have modified this sentence as follows:
“These results strongly suggest that greater treatment effects and even normotension might be achieved the earlier in life the treatment is started.”
Reviewer 3 Report
· Abstract, lines 19-20: the text is confusing. Contradictory results for collagen content and fibrosis are reported whereas increased collagen content is one of the markers of fibrosis.
· Introduction section is too long. The quantitative results obtained in other studies should not be quoted in detail in the Introduction.
· Lines 101-102: when author’s previous studies are reported, the age of “young” and “old” SHR should be specified.
· Lines 116-117: the age at which treatment was started should be specified unequivocally. In the present form it is unclear at which age the experiments were started.
· Line 124: how were tablets administered?
· Materials and methods: it should be clearly stated which data were new and which part of the study is solely based on the re-analysis of previously obtained results.
· Figure 1B: how was treatment effect calculated? Was the difference between pre- and post-treatment BP calculated for each animal and then mean +/- SEM calculated from these data?
· Authors could consider to evaluate treatment efficacy not only according to relative (%) BP reduction but also according to absolute (in mmHg) BP reduction in both age groups.
· The effect on cardiac hypertrophy could be partially mediated by BP reduction and partially being BP-independent (direct antihypertrophic effect). It would be reasonable to calculate the difference in heart weight normalized per unit of BP change to address this issue.
· The significant limitation of this study is that only systolic BP was measured. Full assessment of antihypertensive effectiveness should include the effect on DBP/MBP as well.
· Lines 282-283: the results of this study should not be referred to as effects in the pre-hypertensive stage because young SHR used were already hypertensive.
· Discussion is too long. Many aspects covered in the Discussion are not directly associated with the results such as the advantages of combination therapy because only combination therapy was used by the authors.
ors.
Author Response
We thank this reviewer for his/her valuable comments.
- Abstract, lines 19-20: the text is confusing. Contradictory results for collagen content and fibrosis are reported whereas increased collagen content is one of the markers of fibrosis.
RESPONSE: We have deleted “and collagen expression” from the Abstract. A possible explanation for this contradictory result would be too long for the Abstract, but is discussed in the Discussion section.
- Introduction section is too long. The quantitative results obtained in other studies should not be quoted in detail in the Introduction.
RESPONSE:We have shortened the Introduction section and have deleted many quantitative results quoted from other studies.
- Lines 101-102: when author’s previous studies are reported, the age of “young” and “old” SHR should be specified.
RESPONSE:We inserted the age of the animals: “… young (aged 7-10 weeks) … and old (aged 60-82 weeks) … (SHR) …”
- Materials and methods:Lines 116-117: the age at which treatment was started should be specified unequivocally. In the present form it is unclear at which age the experiments were started.
RESPONSE:This is stated below in Lines 121-123: in ySHR at 7 weeks of age, in oSHR at 60 weeks of age. For more clarity, we have replaced the word “experiment” by “treatment period”.
- Materials and methods:Line 124: how were tablets administered?
RESPONSE:We have added some text into this sentence: “The medication (or placebo in Ctrl animals) was administered as tablets for oral uptake given into the cages along with chow once daily between 9:00 a.m. and 10:00 a.m.”
When the tablets were given into the cages, part of them was eaten immediately, another large part was eaten within the next 6 h. This has been described in Hawlitschek et al., 2022, SJBS [reference #26].
- Materials and methods: it should be clearly stated which data were new and which part of the study is solely based on the re-analysis of previously obtained results.
RESPONSE:All data presented here were obtained from the animal experiments in the previous studies. The main data were re-analyzed using another statistical approach that allows comparing the data from both animal studies. This has been inserted into the text.
- Figure 1B: how was treatment effect calculated? Was the difference between pre- and post-treatment BP calculated for each animal and then mean +/- SEM calculated from these data?
RESPONSE:Only for SBP, we had pre- and post-treatment values. For reasons of consistency, we defined the treatment effect as the difference between the average value of the respective Ctrl group (y or o) and the individual value of the treated animal, normalized to the average yCtrl or oCtrl value (given in %) as described in the Methods section. This definition was applied to all parameters. We have inserted some more details on SBP change over time, in particular, data from the yC+N group. Moreover, we have added a Table containing baseline and final data of SBP, heart rate and body weight of the young and old animal groups.
- Authors could consider to evaluate treatment efficacy not only according to relative (%) BP reduction but also according to absolute (in mmHg) BP reduction in both age groups.
RESPONSE:We have calculated the absolute reduction of SBP and inserted these values into the text. The differences between the absolute values of BP reduction were also not significant.
- The effect on cardiac hypertrophy could be partially mediated by BP reduction and partially being BP-independent (direct antihypertrophic effect). It would be reasonable to calculate the difference in heart weight normalized per unit of BP change to address this issue.
RESPONSE:We thank this reviewer for this interesting proposal. We have calculated these values and inserted them into the text (Results section). In oSHR, the large increase in heart weight is effectively counteracted by direct antihypertrophic effects, which is reflected in a higher HW reduction per unit BP reduction in oSHR compared to ySHR. This has been inserted into the Discussion section (subsection 4.2.2. Effects on cardiac hypertrophy and remodeling in old SHR).
- The significant limitation of this study is that only systolic BP was measured. Full assessment of antihypertensive effectiveness should include the effect on DBP/MBP as well.
RESPONSE:This is indeed a limitation, which is caused by the method. The tail cuff device that we have used did not allow measurement of diastolic BP. We have added a Limitations subsection and inserted this statement there.
- Lines 282-283: the results of this study should not be referred to as effects in the pre-hypertensive stage because young SHR used were already hypertensive.
RESPONSE:This sentence might be misleading. We have modified this sentence as follows:
“These results strongly suggest that greater treatment effects and even normotension might be achieved the earlier in life the treatment is started.”
- Discussion is too Many aspects covered in the Discussion are not directly associated with the results such as the advantages of combination therapy because only combination therapy was used by the authors.
RESPONSE:We have shortened some parts of the Discussion section and focused the discussion to the present results. In particular, we have deleted comparisons of combination therapies with monotherapies.
Reviewer 4 Report
In the present study, Rassler et al. have compared the effectiveness of the antihypertensive treatment with captopril and nifedipine on blood pressure as well as cardiac hypertrophy and fibrosis in both young and old SHR, and found that an early-onset antihypertensive treatment is more effective in reducing systolic blood pressure and in improving cardiac hypertrophy and fibrosis in this model. Because SHR has been regarded as a model of human essential hypertension, the authors’ present findings are of clinical importance and warrant further investigation for clinical translation. However, this reviewer has several concerns that should be adequately addressed.
1. Materials and Methods: Were all SHR derived from the same closed colony in the present study? Please clarify. The SHR has a significant genetic diversity between laboratories.
2. Results: Could the authors please provide the data on systolic blood pressure, heart rate, and body weight before and after treatment in both young and old SHR groups, perhaps in a table?
3. It seems that collagen 1 and collagen 3 are not good biomarkers to diagnose cardiac fibrosis. What is the rational to measure these markers as indicators of cardiac fibrosis? Please explain.
4. I assume the body weight of young and old SHR would be around 250g and 400g, respectively. Perhaps, the doses per body weight of captopril and nifedipine were significantly lower in the old SHR than in young SHR, and thus the amounts of drugs may not be sufficient in old SHR group. Indeed, even after the treatment with captopril and nifedipine, the SBP in old SHR were 194 mmHg which did not reach normotensive WKY levels (around 140mmHg), which might lead to the insufficient reduction of cardiac fibrosis in old SHR.
5. It has been reported that, in rats, aging increases the incidence of cardiac hypertrophy and fibrosis and that ACE inhibitors exert cardioprotective effect independently of blood pressure lowering. Thus, a part of the beneficial effects of captopril and nifedipine could be explained by its specific effects on cardiac hypertrophy and fibrosis associated with aging in old SHR. Accordingly, it would be desirable to test whether the treatment with captopril and nifedipine reduces cardiac hypertrophy and fibrosis in both young and old normotensive WKY rats. Do the authors have any data on this? If not, the authors should discuss this in the Discussion with relevant citations.
6. In the present study, the effect of the treatment with captopril and nifedipine on cardiac function was not tested. This should be mentioned in the Discussion as a limitation of this study.
Author Response
We thank this reviewer for his/her valuable comments.
- Materials and Methods: Were all SHR derived from the same closed colony in the present study? Please clarify. The SHR has a significant genetic diversity between laboratories.
RESPONSE: All animals from both cohorts have been cultured at Charles River (Germany). We have inserted this statement into the Methods section.
- Results: Could the authors please provide the data on systolic blood pressure, heart rate, and body weight before and after treatment in both young and old SHR groups, perhaps in a table?
RESPONSE: We have inserted a Table containing the baseline and the final values of SBP, HR and BW of all animal groups. Both young and old animals were allocated to the treatment groups (Ctrl or C+N) at random; this statement was inserted into the Methods section, too.
- It seems that collagen 1 and collagen 3 are not good biomarkers to diagnose cardiac fibrosis. What is the rational to measure these markers as indicators of cardiac fibrosis? Please explain.
RESPONSE: The rationale to measure collagen mRNA expression was that previous studies from our group showed a good correlation between collagen mRNA expression and collagen amount in the histological specimen (Briest et al., 2001, 2003, references #50,51). Fibrotic processes include enhanced collagen turnover and an increased collagen accumulation. We agree with this reviewer that mRNA expression may not adequately reflect the accumulation of collagen. Moreover, collagen mRNA and protein content develop with aging in a nonsynchronous way [Besse et al., 1994, reference #46]. Analysis of collagen protein concentration or histological evidence of collagen accumulation would more adequately reflect the development of fibrosis. We have discussed this point in the Discussion section (subsection 4.2.2. Effects on cardiac hypertrophy and remodeling in old SHR). Unfortunately, we have not analyzed collagen protein concentration. We have assessed the degree of fibrosis in histological specimens as an expression of the result of the fibrotic processes. We consider the histological degree of fibrosis to be a reliable and meaningful marker of fibrosis. We have added a Limitations subsection and inserted a statement there.
- I assume the body weight of young and old SHR would be around 250g and 400g, respectively. Perhaps, the doses per body weight of captopril and nifedipine were significantly lower in the old SHR than in young SHR, and thus the amounts of drugs may not be sufficient in old SHR group. Indeed, even after the treatment with captopril and nifedipine, the SBP in old SHR were 194 mmHg which did not reach normotensive WKY levels (around 140mmHg), which might lead to the insufficient reduction of cardiac fibrosis in old SHR.
RESPONSE: The doses were defined as mg per kg per day, this is stated in the Methods section. This means, the doses were adjusted to the individual body weight of each animal. In addition, the animals were weighed regularly (every 1-2 weeks), and the drug dose was also adjusted to changes in individual body weight.
- It has been reported that, in rats,aging increases the incidence of cardiac hypertrophy and fibrosis and that ACE inhibitors exert cardioprotective effect independently of blood pressure lowering. Thus, a part of the beneficial effects of captopril and nifedipine could be explained by its specific effects on cardiac hypertrophy and fibrosis associated with aging in old SHR. Accordingly, it would be desirable to test whether the treatment with captopril and nifedipine reduces cardiac hypertrophy and fibrosis in both young and old normotensive WKY rats. Do the authors have any data on this? If not, the authors should discuss this in the Discussion with relevant citations.
RESPONSE: Both drugs, captopril and nifedipine, exert direct antihypertrophic effects on cardiomyocytes. This has been inserted into the Discussion section. However, we have no own data on the effects of these drugs on WKY rats. In the original study on young rats, we had a group of WKY rats as normotensive control, but these animals did not receive any therapy. In the old SHR study, we had not investigated old WKY rats. In a study of Demirci et al. (2005; reference #24 in the manuscript) young, adult and old SHR and WKY were treated with losartan, another antagonist of the RAS. With this treatment, only in old but not in young or adult WKY rats, significant reduction of cardiac hypertrophy was achieved. LV contractility and relaxation, which may reflect increased stiffness of the myocardium, and aortic contractility and relaxation were significantly improved with losartan in WKY rats of all three levels of age. We have mentioned these results in the Discussion section (subsections 4.1. Effects on SBP and 4.2.2. Effects on cardiac hypertrophy and remodeling in old SHR)
- In the present study, the effect of the treatment with captopril and nifedipine on cardiac function was not tested. This should be mentioned in the Discussion as a limitation of this study.
RESPONSE: In the Discussion section, we have discussed our results by comparing the changes on tissue level with functional changes observed in other studies (e.g., echocardiographic studies). In addition, we have added a Limitations section and stated the lack of functional measurements as an important limitation to this study.
Round 2
Reviewer 2 Report
No new comments.